# Multicentre, non-interventional study to assess the profile of patients with uncontrolled rhinitis prescribed a novel formulation of azelastine hydrochloride and fluticasone propionate in a single spray in routine clinical practice in the UK

Glenis Scadding,[1] David Price,[2,3] Tariq El-Shanawany,[4] Shahzada Ahmed,[5] Jaydip Ray,[6] Ravishankar Sargur,[7] Nirmal Kumar[8]

For numbered affiliations see end of article.

Correspondence to
Dr Glenis Scadding;
g.scadding@ucl.ac.uk

## ABSTRACT

**Objective:** The aims of this study were (1) to characterise the type of patient prescribed MP-AzeFlu (Dymista, a novel formulation of azelastine hydrochloride, fluticasone propionate and excipients in a single spray) in real life in the UK and physicians' reasons for prescribing it and (2) to quantify the personal and societal burden of allergic rhinitis (AR) in the UK prior to MP-AzeFlu prescription.

**Design, setting and participants:** This multicentre, non-interventional study enrolled patients (n=193) with moderate-to-severe AR and acute symptoms who were eligible to receive treatment with MP-AzeFlu according to its licensed indications. Information was gathered on patient demographics, AR history and symptom severity, symptomatology and AR treatments in the previous calendar year (prior to MP-AzeFlu prescription). Physicians also recorded the number of previous AR visits, specific reasons for these visits and their reason for prescribing MP-AzeFlu.

**Results:** Most patients had seasonal AR either alone (10.4%) or in combination with perennial AR (35.2%), but many had AR of unknown origin (35.8%). Prior to MP-AzeFlu prescription, patients reported troublesome symptoms (78.2%) and sleep disturbance (64.8%), with congestion considered the most bothersome (54.4%) and ocular symptoms reported by 68.4% of patients. The most frequent reason for MP-AzeFlu prescription was that other therapies were not sufficient in the past (78.8%) or not sufficient to treat acute symptoms (16.1%). 79.3% of patients reported using ≥2 AR therapies in the past year. An average of 1.6 (SD 1.9) doctor visits due to AR were reported prior to MP-AzeFlu prescription.

**Conclusions:** In the UK, MP-AzeFlu was prescribed for individuals (≥12 years) with moderate/severe AR

## Strengths and limitations of this study

- Sufficiently large patient numbers to draw general conclusion on use of MP-AzeFlu in routine clinical practice in the UK.
- Low amount of missing data.
- Real-life data obtained from daily practice highlighting the burden of allergic rhinitis in the UK and the type of patient prescribed MP-AzeFlu.
- Relates type of patient prescribed MP-AzeFlu with the labelled indication, thus providing sound estimates for relevant subgroups from a socioeconomic perspective.
- For such single-arm epidemiological research, standard bias limiting methods (blinding, randomisation) could not be applied.

irrespective of (1) previous AR treatment (mono or multiple), (2) previous or likely treatment failure, (3) phenotype, (4) number of previous physician visits for AR and (5) for the relief of both acute symptoms and in anticipation of allergen exposure.

## INTRODUCTION

One in four individuals in the UK has allergic rhinitis (AR),[1,2] yet this disease continues to be underestimated, both in terms of its impact on individuals and their families,[3,4] as well as its cost to the economy. When symptomatic, patients feel the impact of AR on all areas of their daily lives, at home, socially and at work/school.[5-7] Indeed, in

the UK, children with symptomatic AR are more likely to unexpectedly drop an examination grade during summer tests than their non-AR counterparts, and are even more at risk of so doing if taking sedating antihistamines.[8] Poorly controlled AR also has a negative impact on asthma control, equivalent to that induced by smoking.[9] While the direct costs (eg, drug costs) associated with AR are relatively low, the indirect costs due to absenteeism and presenteeism are disproportionately large, estimated at £1.4 billion/year in the UK alone.[10]

The burden of AR and the importance of achieving disease control has now been recognised both by the European Academy of Allergy and Clinical Immunology[11] and at the European Union (EU) level.[12] [13] Provision of more effective and fast acting AR treatments proven to achieve guideline-defined control in real life should help reduce the burden of disease and associated management costs.[14]

MP-AzeFlu (Mylan Inc, Canonsburg, PA, USA) comprises a novel formulation containing an intranasal antihistamine (azelastine hydrochloride), an intranasal corticosteroid (INS), fluticasone propionate (FP) and excipients delivered in a single spray. It has been approved for use in the UK since January 2013, indicated for the relief of symptoms of moderate-to-severe seasonal AR (SAR) and perennial AR (PAR) if monotherapy with either intranasal antihistamine or glucocorticoid is not considered sufficient.[15] In a randomised controlled clinical trial (RCT) setting, MP-AzeFlu provided twice the overall nasal and ocular symptom relief as FP or azelastine monotherapy, and provided faster complete (or near-complete) relief in more patients.[16] Patients treated with MP-AzeFlu in real-life settings in Germany, Sweden, Denmark, Norway and Romania experienced rapid and sustained symptom control, with one in two patients reporting that their AR was well controlled after 3 days.[17] In those studies, control was assessed using the a simple visual analogue scale (VAS) score, as endorsed by MACVIA-ARIA (Contre les Maladie Chronique pour un Vieillissement Actif-Allergic Rhinitis and its Impact on Asthma) and incorporated into the updated guideline, called AR clinical decision support system (CDSS).[14] On average, patients treated with MP-AzeFlu crossed the well-controlled VAS score cut-off (ie, 5/10 cm) prior to day 3, irrespective of disease severity, phenotype (SAR, PAR, SAR+PAR and unknown) or previous treatment (with monotherapy or multiple therapies).[17]

Characteristics of patients who received MP-AzeFlu in an RCT setting are well established, due to strict inclusion/exclusion criteria applied to ensure a homogeneous patient population and low external variability.[18] However, detailed characteristics of those patients prescribed MP-AzeFlu in routine clinical practice are unknown. The aim of this study was to (1) characterise the type of patient prescribed MP-AzeFlu in real life in the UK and physicians' reasons for prescribing it; (2) quantify the burden of AR prior to MP-AzeFlu

prescription; and (3) describe AR treatment patterns for these patients prior to MP-AzeFlu use in the UK. The effectiveness of MP-AzeFlu in a real-life pan-European clinical setting has already been established.[17]

## METHODS
### Study design
This was a multicentre, non-interventional study conducted in the UK between October 2013 and May 2014, as part of a larger Europe-wide study of similar design. During the study, patients were treated with MP-AzeFlu (one spray in each nostril twice daily). MP-AzeFlu is indicated for the relief of symptoms of moderate-to-severe SAR and PAR if monotherapy with either intranasal antihistamine or glucocorticoid is not considered sufficient.[15] The study was performed in line with the current UK laws and guidelines. The study was registered at the European Network of Centres for Pharmacoepidemiology and Pharmacovigilance.

### Patients
Patients were eligible for inclusion into the study if they were aged ≥12 years, had a documented diagnosis of moderate-to-severe SAR or PAR and for whom monotherapy with either intranasal antihistamine or glucocorticoid were not considered sufficient.[15] Patients were required to have acute AR symptoms on the study inclusion day. Patients with SAR were defined as those with a documented allergy to at least one pollen allergen (ie, spring, summer and/or autumn pollen) but no non-pollen allergens. Patients with PAR were those with a documented allergy to at least one non-pollen allergen (ie, dust mites, pet dander and/or mould) but no pollen allergens. Patients with both SAR+PAR had documented allergy to at least one pollen and at least one non-pollen allergen. Rhinitis of unknown origin was defined as allergy to other or unknown allergens (ie, not one of the allergens listed above) or unknown allergens (ie, rhinitis indicated from history but no specific IgE data).

Patients were excluded if they had hypersensitivity to azelastine hydrochloride, FP or any MP-AzeFlu excipients. There were no restrictions regarding concomitant treatments, apart from ritonavir, which was to be avoided. Female patients were excluded if they were pregnant or breast feeding. All patients provided written informed consent prior to their participation in the study. If patients were below 16 years old, their caregiver also provided written informed consent.

### Physicians
Physicians who were usually involved in AR management participated in this study and included secondary care ear-nose-and-throat (ENT) specialists, immunologists and allergists. These physicians were identified by an independent agency (pH Associates, UK). Each physician could enrol up to 20 patients. The decision

whether to include a patient in the study was made by the physician independently from and after the decision to prescribe MP-AzeFlu had been made.

## MP-AzeFlu use in routine clinical practice

Information on patient demographics, clinical symptoms and AR treatments in the previous calendar year (prior to MP-AzeFlu prescription) was documented by physicians. Physicians also recorded information on type of AR, number of visits in the current calendar year due to AR, predominant symptoms and ARIA-defined AR severity. The reason for the patient's visit (acute AR symptoms, expected allergen exposure in near future or other) and the reason for prescribing MP-AzeFlu (other therapies were not sufficient in the past, other therapies are not considered to be sufficient to treat acute symptoms or other) was documented by the physician. All data were recorded by physicians in a prespecified English language electronic case report from (Trium Analysis Online GmbH) based on the investigators' patient files and on information obtained at the patient visit.

## Statistics

This study included 193 patients with AR, which was deemed sufficient to provide insight into use of MP-AzeFlu in routine clinical practice in the UK. Analyses were conducted on the safety population, defined as all patients who were treated at least once with MP-AzeFlu and whose physician provided an electronic signature to confirm data accuracy. All data were reported with descriptive statistics. Categorical data were described in terms of frequency and as percentage of patients. Continuous data were described as mean and SD. Analyses were performed by the Contract Research Organisation, Syneed Medidata GmbH, with SAS V.9.1.3.

## RESULTS

### Patient disposition

The study was conducted in 15 study centres in the UK who enrolled 195 patients. Of these, 2 patients were excluded from data analysis due to unconfirmed data documentation, leaving 193 confirmed patients in the safety population.

### Patient demographics and baseline characteristics

A summary of patient demographics and baseline clinical characteristics is presented in table 1. There were slightly more women than men in the safety population. The mean (SD) age of the study population was 37.6 (16.9) years, with most of the patients aged 18–65 years. Almost half of patients had SAR, either alone or in combination with PAR. A relatively high proportion of patients had rhinitis with aetiology of unknown origin (table 1). The vast majority of patients had confirmed ARIA-defined moderate-to-severe disease. AR symptoms were troublesome to patients, negatively impacting

**Table 1** Patient demographics and baseline clinical characteristics (N=193)

| Characteristic | n (%) |
|---|---|
| Gender | |
| Female | 107 (55.4) |
| Age, years | |
| 12–17 | 22 (11.4) |
| 18–65 | 155 (80.3) |
| >65 | 16 (8.3) |
| **Duration of rhinitis,*** year (mean (SD)) | 8.5 (9.4) |
| **Type of rhinitis** | |
| SAR | 20 (10.4) |
| PAR | 36 (18.6) |
| SAR+PAR | 68 (35.2) |
| Unknown origin | 69 (35.8) |
| **Severity of AR criteria†** | |
| Troublesome symptoms | 151 (78.2) |
| Sleep disturbance | 125 (64.8) |
| Impairment of daily activities/leisure/sport | 109 (56.5) |
| Impairment of school/work | 93 (48.2) |
| At least one criteria | 175 (90.7) |
| **Predominant symptoms** | |
| Nasal congestion | 105 (54.4) |
| Rhinorrhoea | 26 (13.5) |
| Sneezing | 20 (10.4) |
| Nasal itching | 10 (5.2) |
| Unknown | 32 (16.6) |
| **Concomitant ocular symptoms** | 132 (68.4) |

*N=136.
†Moderate/severe AR if at least one criterion was met.
AR, allergic rhinitis; PAR, perennial allergic rhinitis; SAR, seasonal allergic rhinitis.

sleep, daily activities, leisure and/or sport and school/work. Nasal congestion was by far the most common predominant symptom. Ocular symptoms were present in over two-thirds of patients (table 1).

### Physician visits

In the current calendar year, the mean (SD) number of physician visits due to AR was 1.6 (1.9). In total, 62.7% (n=121) of patients had visited their physician at least once before inclusion into the study in the current calendar year due to their AR (figure 1); with most attending once or twice before. However, 23.8% of patients (n=46) had made three or more visits prior to the current visit (figure 1).

### Reasons for patient visit and MP-AzeFlu prescription

MP-AzeFlu was prescribed according to its licensed indication. The most frequent reasons for the physician visit were 'acute AR symptoms' (n=142; 73.6%), 'expected allergen exposure in the near future' (n=21; 10.9%) and 'other' (n=37; 19.2%). The most frequent reason for prescribing MP-AzeFlu was that 'other therapies were not sufficient in the past'. For the remaining patients, other reasons were cited, including 'other therapies were not considered sufficient to treat acute symptoms' (figure 2).

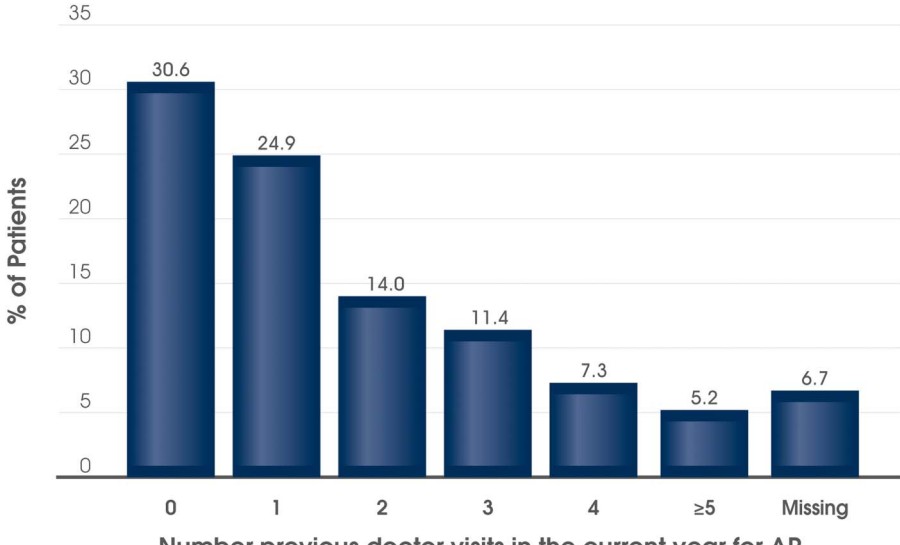

**Figure 1** Number of previous physician visits for AR in the current calendar year prior to prescribing MP-AzeFlu. N=193 patients with moderate-to-severe AR attending routine clinical practice in the UK. AR, allergic rhinitis.

## AR treatments in the last year

The majority of patients had received or used at least one AR therapy, either with prescription or over the counter, prior to MP-AzeFlu prescription (table 2). The AR medications most frequently used in the previous year were INS, oral antihistamines and oral/intranasal decongestants. Eye drops, either antihistamine or mast cell stabiliser use, was reported by over a quarter of patients. Systemic corticosteroid use was noted for quite a high proportion of patients (table 2). The majority of patients reported using multiple AR treatments in the previous year, prior to MP-AzeFlu use; dual AR therapy was most commonly reported, but many also used triple and even quadruple AR therapies (figure 3). A total of 25 patients (13.0%) reported previously using five or more AR treatments prior to MP-AzeFlu prescription. Seven patients (3.6%) stated that they were undergoing immunotherapy at the time of the inclusion visit and 1 (0.5%) had received an immunotherapy course in the previous calendar year.

## DISCUSSION

This study represents a valuable addition to the AR knowledge base for several reasons. First, it provides important epidemiological data on the type of patient with AR presenting to secondary care allergy clinics in the UK, as well as information on AR history and symptomatology of those who received MP-AzeFlu. Second, it confirms the enormous (and frequently underestimated)[4] burden imposed by uncontrolled AR symptoms on patients' lives. Third, it provides information on AR treatment pattern in the UK (prior to MP-AzeFlu prescription) and the shortcomings of commonly used treatments in providing symptom control. It also explores the high socioeconomic burden imposed by AR, as seen through medication usage and number of doctor visits for AR. Finally, this study details how the indication for MP-AzeFlu,[15] a relatively new addition to the AR armoury, was interpreted by prescribing physicians in secondary and tertiary care during the first months of its introduction and use in the UK.

**Figure 2** Most frequent reasons reported by physicians for prescribing MP-AzeFlu to patients with moderate-to-severe allergic rhinitis (n=193) attending routine clinical practice in the UK. Tx, treatment.

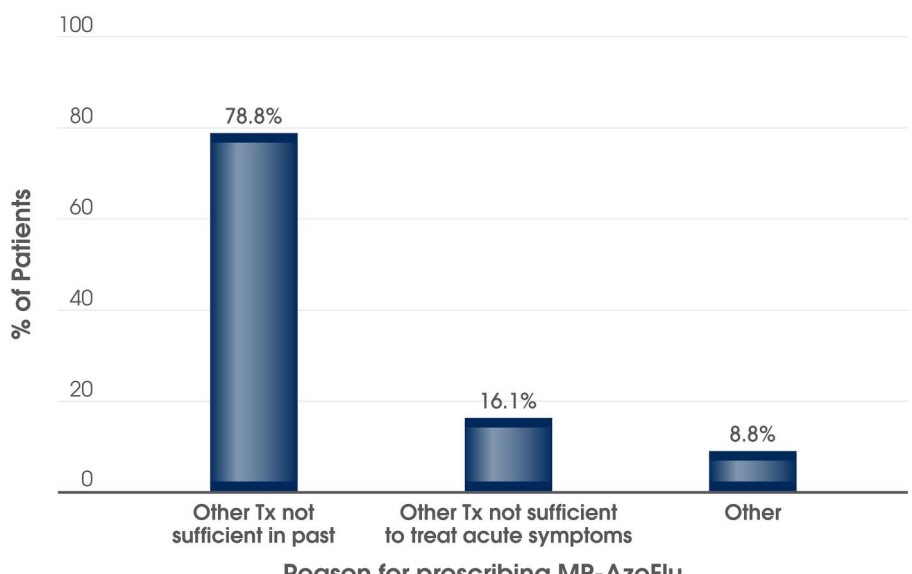

**Table 2** AR treatments in the past year (N=193)

| AR treatments (multiple entries possible) | n (%) |
|---|---|
| Intranasal corticosteroid | 162 (83.9) |
| Oral antihistamine | 128 (66.3) |
| Intranasal decongestant | 47 (24.4) |
| Oral decongestant | 41 (21.2) |
| Systemic corticosteroid | 36 (18.7) |
| Intranasal antihistamine | 31 (16.1) |
| Ocular antihistamine | 31 (16.1) |
| Ocular mast cell stabiliser | 23 (11.9) |
| Oral leucotriene receptor antagonist | 12 (6.2) |
| Intranasal mast cell stabiliser | 1 (0.5) |
| Any other | 11 (5.7) |
| Unknown | 8 (4.2) |
| None | 3 (1.6) |
| Immunotherapy (in past or ongoing) | 8 (4.1) |
| Multiple therapies (excluding immunotherapy) | 153 (79.3) |
| AR, allergic rhinitis. | |

Physicians who took part in this non-interventional study (ie, allergists, ENT specialists and immunologists) prescribed MP-AzeFlu to adolescents, adults and elderly patients with moderate/severe AR (ie, ≥1 ARIA severity criteria)[19] in line with its indication.[15] It was prescribed to those patients for whom other AR therapies were not considered sufficient, and also to those for whom other therapy had not worked in the past. This decision depends very much on physician and patient preference; that is, the preference for rapid AR symptom control straight away with the most effective pharmacological treatment option, or a gradual therapy step up until control has been achieved. MP-AzeFlu was prescribed for the treatment of acute symptoms, and also prophylactically in the anticipation of allergen exposure in the near future. It was prescribed for patients with SAR, PAR, both SAR and PAR and for those whose AR aetiology was not defined, which represented about one-third of patients in the current study. This unknown group is likely quite heterogeneous in nature, and could include those with non-AR, or local AR, as well as those with SAR or PAR of unknown allergen or allergen not listed. Finally, MP-AzeFlu was prescribed for patients previously on AR monotherapy and for those attending clinic for the first time in the season. It was also prescribed for patients with more complicated disease who had attended many times in the past year for AR, and for those on multiple therapies or immunotherapy.

Although MACVIA ARIA clearly recommends MP-AzeFlu for the treatment of intermittent and persistent AR,[14] a significant proportion of patients who may benefit from this treatment in the UK have not received it. The need for wider awareness and use of MP-AzeFlu in the UK (particularly in primary care) is shown by the fact that prior to MP-AzeFlu prescription the majority of patients reported troublesome symptoms, sleep disturbance and impairment of their daily activities, with almost half acknowledging a negative impact of their AR symptoms on performance at work or school. The negative impact of AR symptoms on daily activities has previously been shown to be greater for patients with AR than those with diabetes mellitus or hypertension.[20] Others have shown that living with symptomatic AR can impair school performance,[8] impair driving ability to the same extent as a blood alcohol level of 0.05%,[21] reduce work productivity[10 22] and induce absenteeism.[10] The latter two carry a hefty price tag, with absenteeism due to AR estimated to cost the UK economy £1.14 billion/year[10] and the cost of lost productivity due to AR thought to exceed that of chronic heart

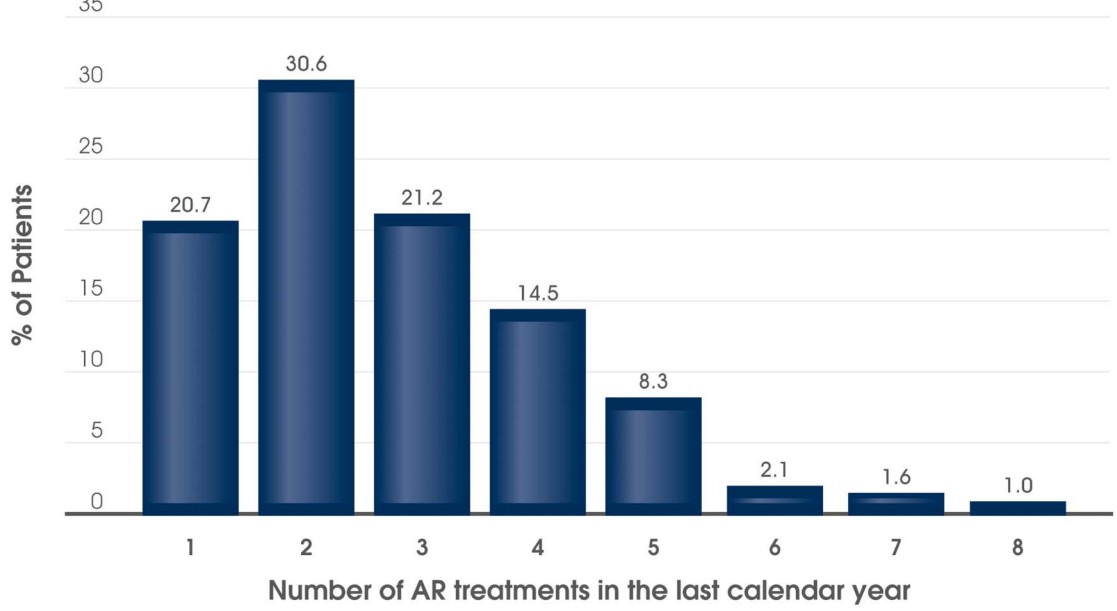

**Figure 3** Number of AR treatments (prescription and over the counter) used in the last calendar year by patients with AR with moderate-to-severe AR (n=193) attending routine clinical practice in the UK. AR, allergic rhinitis.

disease, asthma, diabetes, hypertension and respiratory infections combined![22]

The results of this study also show that the most commonly prescribed AR treatments in the UK do not provide adequate symptom relief for many patients, further endorsing the need for MP-AzeFlu. Prescribing physicians considered that other therapies were not sufficient in the past for 66.4% of patients and would not provide adequate symptom relief for 34.5% of patients. Most patients prescribed MP-AzeFlu had previously been treated with INS and/or oral antihistamines. Inadequacy of previous AR treatments was also evident from the high symptomatic burden on treatment reported by patients, the high degree of polypharmacy noted (79.3%) and the high incidence of use of systemic corticosteroids (18.7%) and decongestants (45.6%). The former are rarely indicated for the management of rhinitis (except for severe obstruction or short-term rescue use),[23] while long-term use of the latter has been associated with rhinitis medicamentosa. A previous survey conducted in the UK also revealed that patients retained a high nasal and ocular symptom burden despite treatment with monotherapy or multiple AR therapies.[10] Furthermore, a retrospective observational study with data from the Optimum Patient Care Research Database showed that in the UK monotherapy with antihistamines or INS provided insufficient symptom relief for about 20% of patients with SAR.[24] AR monotherapy was prescribed first for most patients, but by season end 45% of patients with SAR had received multiple therapy, a practice which is not recommended by ARIA[19] due to insufficient evidence.[25][26] The current British Society for Allergy and Clinical Immunology (BSACI) UK guidelines suggest that additional therapies should be considered for brief-targeted symptom relief only.[23]

The concept of achieving symptom relief (ie, reduction in symptom severity) has been now superseded by the concept of achieving symptom control.[14] A simple VAS is recommended by MACVIA ARIA as the new language of AR control.[12] The VAS has been used extensively in the field of AR to assess symptom severity[27] and treatment effectiveness,[28] to define phenotypes, such as severe chronic upper airway disease,[29] and to define clinically relevant effect.[30] A VAS score of 5/10 cm is the cut-off used to define control and guide treatment decisions as part of an updated ARIA AR treatment algorithm called the AR CDSS.[14] It has also been incorporated into an app (*Allergy* Diary) to empower patients to take control of their own AR.[12] Data obtained from several countries (Germany, Sweden, Norway, Denmark and Romania) which gathered real-life data as part of a multinational pan-European study have shown that on average patients treated with MP-AzeFlu achieve this AR CDSS VAS score cut-off (ie, 5 cm) prior to day 3, with one in two patients perceiving their AR as well controlled at that time point.[31–35] The UK data presented here are taken from the same multinational pan-European study. Although, effectiveness data were not recorded in the UK arm,

results are expected to be consistent with those found in other countries with similar patient characteristics. MP-AzeFlu has the potential to reduce costs associated with managing AR in the UK, by reducing multiple prescriptions, number of doctor visits, referrals to secondary care and potentially reducing the number of patients referred for allergen-specific immunotherapy (AIT); supporting UK National Health Service (NHS) cost-saving initiatives to manage more chronic conditions in the primary care setting.

The limitations of this study are those typically associated with non-interventional, observational data sets as the standard methods of blinding and randomisation cannot be applied. Thus, confounding can never safely be ruled out. Another limitation is that our study included no investigation of the effectiveness of MP-AzeFlu, and there was no direct head-to-head comparison of MP-AzeFlu with other AR treatments. However, the effectiveness of MP-AzeFlu has been demonstrated in real-life studies conducted in other countries.[17] The strength of the evidence is derived from the inclusion of a sufficiently large patient population to draw general conclusion on use and safety of MP-AzeFlu in routine clinical practice in UK. The type of patient prescribed MP-AzeFlu in the UK was consistent with the patient profile identified in other countries including Germany, Sweden, Denmark and Romania, supporting the generalisability of our data to a wider pan-European population. In the pooled analysis for all countries as well as for the presented UK data, the study data were complete for most demographic variables; therefore, missing data are not considered as a potential source of bias for the study results.

In conclusion, the symptomatic burden of AR remains high for many patients in the UK, who live with uncontrolled disease, despite treatment with monotherapy and multiple therapies and repeat doctor visits. In the UK, MP-AzeFlu was prescribed for individuals (≥12 years) with moderate/severe AR irrespective of (1) previous AR treatment (mono or multiple), (2) previous or likely treatment failure, (3) phenotype, (4) number of previous physician visits for AR and (5) for the relief of both acute symptoms and in anticipation of allergen exposure. A more effective treatment option, like MP-AzeFlu, with established superiority over INS and antihistamine monotherapy, should improve AR control.

**Author affiliations**
[1]Royal National Throat, Nose and Ear Hospital, London, UK
[2]University of Aberdeen, Aberdeen, Scotland
[3]Observational and Pragmatic Research Institute Pte Ltd, Singapore, Singapore
[4]Univeristy Hospital of Wales, Cardiff, UK
[5]Queen Elizabeth Hospital, Birmingham, UK
[6]Royal Hallamshire, Sheffield, UK
[7]Northern General Hospital, Sheffield, UK
[8]Royal Albert & Edward Hospital, Wigan, UK

**Acknowledgements** The authors thank the following investigators for their participation in the study: Carl Philpott, James Paget Hospital, Great Yarmouth; Dr Dermot Dalton, North Devon District Hospital, Barnstable; Dr Sadie Khawaja, Stepping Hill Hospital, Stockport; Dr Tom Dawson, Worcester Royal Hospital, West Midlands; Alagar Chandra-Mohan, Ysbyty Gwynedd Hospital, Wrexham, Wales; Arul Puveendran, Sunderland Royal Hospital, Sunderland; Paul Counter, Cumberland Infirmary, Carlisle; Ramsam Ullah, Royal Victoria Hospital, Belfast, Northern Ireland; San Sunkaraneni, Royal Surrey Hospital, Guildford; Professor Chris Raine, Bradford Royal Infirmary, Bradford. The authors also gratefully acknowledge each of the clinical research facilities which took part in this study. Medical writing assistance in the preparation of this manuscript was provided by Dr Ruth Murray and Dr Winnie McFadzaen, Medscript, funded by MEDA Pharma GmbH & Co KG. The authors thank Dr DunTung Nguyen and Dr Hans-Christian Kuhl for critical assessment of the data and manuscript review.

**Funding** This work was supported by MEDA Pharma GmbH & Co KG.

**Competing interests** GS has received fees for advising Meda and for speaking at meetings on Dymista. DP has received fees for advising Meda, research grants from Meda for studies on allergic rhinitis and fees for speaking at meetings and symposia supported by Meda. TE-S has received educational support, speaker fees and/or advisory board fees from Meda, Allergy Therapeutics and ALK. SA has received fees for speaking at a meeting by Meda. JR has received fees for speaking at a meeting by Meda. RS has received honorarium in Meda sponsored educational events. NK has received fees for advising and for speaking at meetings arranged by Meda.

**Patient consent** Obtained.

**Ethics approval** National Research Ethics Service Committee London—Brent.

**Provenance and peer review** Not commissioned; externally peer reviewed.

**Data sharing statement** No additional data are available.

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
