## [Reviewer comments · BMJ Open]

ARTICLE DETAILS

TITLE (PROVISIONAL)	Multi-centre, non-interventional study to assess the profile of patients with uncontrolled rhinitis prescribed a novel formulation of azelastine hydrochloride and fluticasone propionate in a single spray in routine clinical practice in the UK
AUTHORS	Scadding, Glenis; Price, David; El-Shanawany, Tariq; Ahmed, Shahzada; Ray, Jaydip; Sargur, Ravishankar; Kumar, Nirmal

VERSION 1 - REVIEW

REVIEWER	Professor Ludger Klimek MD, PhD Center for Rhinology and Allergology Wiesbaden, Germany
REVIEW RETURNED	30-Oct-2016

GENERAL COMMENTS	Important data for real-life treatment of allergic rhinitis patients in the UK. Quality of care of patients should improve, if sufficient data is available on the use of second-line drugs such as MP-AzeFlu. This data should be urgently published.
--

REVIEWER	Giovanni Passalacqua Allergy and Respiratory Diseases University of Genoa, Genoa ITALY
REVIEW RETURNED	11-Nov-2016

GENERAL COMMENTS	GENERAL COMMENT This article describes the results of a physician-based survey in patients with moderate-severe AR in UK, with a particular attention paid at the characteristics of rhinitis before prescribing the new nasal spray (azelastine/fluticasone AZEFLU), and the reasons for prescribing it once introduced as commercial product. The study, described as prospective, non interventional and observational, involved 193 patients with "AR". In the previous calendar year most patients (about 80%) reported severe symptoms, being nasal obstruction the most relevant one, and 65% had also sleep impairment. On average, 2 different treatments had been used, and on average 1.6 physician visits had been made in the previous year. According to physicians' judgement, AzeFlu was prescribed mainly because the previously used therapies were not sufficient to achieve symptom control. The study maybe of interest, since it provides a cross sectional picture of the burden of rhinitis in a real-life setting, and describes the attitude of physicians toward a newly introduced treatment. Nonetheless, in my opinion, there are some major methodological
--

	flaws, some imprecisions in the descriptions of findings, and I have doubts on the possible generalization of results. Finally, as clearly acknowledged by Authors, there is no mention on how the use of the new intranasal product could have changed the clinical outcome. SPECIFIC COMMENTS Page 4, line 16 STRENGTHS and LIMITATIONS OF THIS STUDY first sentence "...confirms the enormous burden imposed by uncontrolled AR symptoms on patients' lives." Since a quite detailed assessment of visits and medication usage (previous year), could it be possible to provide an approximate economical estimate on these parameters? Page 7, line 44. I think that the definition of "AR of unknown origin was defined as allergy to other or unknown allergens (i.e. not one of the allergens listed above) or unknown allergens (i.e. rhinitis indicated from history but not specific IgE data" is too vague to include all these forms under the definition of AR. AR needs the mandatory demonstration of an IgE-mediated mechanism. Thus, including all patients under the umbrella of AR is not scientifically correct in this context. Needs justification. At this point, I should suppose that the definition of PAR, SAR, SAR+PAR were supported by an IgE sensitization diagnosis, but this is not clearly described in the text. Were skin test and/or IgE assay performed in all patients? Page 8, line 37. MP-AzeFlu use in clinical practice. First sentence "Information on patient demographics, clinical symptoms, and previous AR treatments in the last calendar year (prior to MP-AzeFlu prescription) was documented by physicians". It is not clear if the study was truly prospective or if data were collected also retrospectively. If this is a prospective study, the Authors must clearly state that patients were enrolled starting 1 year prior to the prescription of the product. If not, the study is not prospective. How were data collected? Agreed database? Questionnaire? Interviews? Clinical records' review? Please specify this aspect. How were the involved physicians enrolled? How were they distributed (tertiary/secondary, specialists/GPs etc) Another formal problem is that rhinitis is classified for severity according to ARIA criteria, but for duration according to the classic distribution (PAR/SAR). According to Fig 1, one third of patients had no physician visit in the previous year. How was the drug prescription assessed in those patients?
--	--

REVIEWER	Caral Irani Hotel Dieu de France hospital, St Joseph University, Beirut, Lebanon
REVIEW RETURNED	26-Nov-2016

GENERAL COMMENTS	The authors have as main objective to see the control of AR prior to Azelastine. The results for that regard are clear. Discussion of "Dymista" reasons for prescription need to be better explored and discussed. It seems that the study could have been done even
--

	without mentioning the introduction of "Dymista"
--	--

REVIEWER	AYSE BACCIOGLU KIRIKKALE UNIVERSITY FACULTY OF MEDICINE, DEPARTMENT OF PULMONARY DISEASES, DIVISION OF IMMUNOLOGY AND ALLERGY KIRIKKALE, TURKEY
REVIEW RETURNED	13-Dec-2016

GENERAL COMMENTS	 1. There are some minor spelling mistakes. For example; abstract- Conclusion: many patients, introduction: line6-...(AR),1 2.... 2. Methods: detailed prescription about MP-AzeFlu may be added (the duration, indications, adverse events....). 3. Study design is confusing. Even it is prospective, all the data is collected retrospectively. It was written that "patients were treated with MP-AzeFlu". So did all patients use MP-azeFlu? Where are the results of the drug use? Whom were most recovered group? 4. Is this a company sponsored study? 5. As I understand this is not a study showing the effectiveness of drug, so we don't know if MP-AzeFlu is more effective than nasal steroid or oral antihistamine. So the last sentence of the conclusion "A more effective treatment option, like MP-AzeFlu, should improve AR control and reduce costs associated with its management" is only an assumption, and should be discarded or replaced to a mild compliment about MP-AzeFlu. 6. Some results comparing azelastine versus nasal steroid in AR/NAR and the for a combination drug may be added (Kalpaklıoğlu, A.F., Baççioğlu Kavut, A., "Comparison of azelastine versus triamcinolone nasal spray in allergic and nonallergic rhinitis", Am J Rhinol Allergy, 24, 29-33 (2010).
--

VERSION 1 – AUTHOR RESPONSE

Reviewer: 1

Reviewer Name: Professor Ludger Klimek MD, PhD

Institution and Country: Center for Rhinology and Allergology, Wiesbaden, Germany

Please state any competing interests: None declared

Important data for real-life treatment of allergic rhinitis patients in the UK. Quality of care of patients should improve, if sufficient data is available on the use of second-line drugs such as MP-AzeFlu.

This data should be urgently published.

Response: Many thanks for your kind comments

Reviewer: 2

Reviewer Name: Giovanni Passalacqua

Institution and Country: Allergy and Respiratory Diseases, University of Genoa, Genoa, ITALY

Please state any competing interests: None to declare

GENERAL COMMENT

This article describes the results of a physician-based survey in patients with moderate-severe AR in UK, with a particular attention paid at the characteristics of rhinitis before prescribing the new nasal spray (azelastine/fluticasone AZEFLU), and the reasons for prescribing it once introduced as commercial product.

The study, described as prospective, non-interventional and observational, involved 193 patients with “AR”. In the previous calendar year most patients (about 80%) reported severe symptoms, being nasal obstruction the most relevant one, and 65% had also sleep impairment. On average, 2 different treatments had been used, and on average 1.6 physician visits had been made in the previous year. According to physicians’ judgement, AzeFlu was prescribed mainly because the previously used therapies were not sufficient to achieve symptom control.

The study may be of interest, since it provides a cross sectional picture of the burden of rhinitis in a real-life setting, and describes the attitude of physicians toward a newly introduced treatment. Nonetheless, in my opinion, there are some major methodological flaws, some imprecisions in the descriptions of findings, and I have doubts on the possible generalization of results. Finally, as clearly acknowledged by Authors, there is no mention on how the use of the new intranasal product could have changed the clinical outcome.

Response: Thank you for your comments. We have addressed methodological strengths and limitations directly following the abstract (page 3, line 54) and included a generalizability statement in the discussion section (page 19, line 325). Although assessment of MP-AzeFlu was not permitted in this UK arm of study NIS 3299, it was assessed in 5 other countries using a simple VAS1 – the ARIA-endorsed tool for monitoring of AR control.² Bearing in mind that the type of patient prescribed MP-AzeFlu in these countries was similar to the UK, one would expect the effectiveness of MP-AzeFlu in UK patients to be similar to that observed in patients from Germany, Sweden, Norway, Denmark and Romania. Indeed, VAS scores before and after MP-AzeFlu treatment showed little inter-country variability.¹

SPECIFIC COMMENTS

Page 4, line 16 STRENGTHS and LIMITATIONS OF THIS STUDY first sentence “...confirms the enormous burden imposed by uncontrolled AR symptoms on patients’ lives.” Since a quite detailed assessment of visits and medication usage (previous year), could it be possible to provide an approximate economical estimate on these parameters?

Response: Thank you. As per editorial comments we have amended the strength and limitations of this study to focus on methodological strengths and limitations.

Page 7, line 44. I think that the definition of “AR of unknown origin was defined as allergy to other or unknown allergens (i.e. not one of the allergens listed above) or unknown allergens (i.e. rhinitis indicated from history but not specific IgE data” is too vague to include all these forms under the definition of AR. AR needs the mandatory demonstration of an IgE-mediated mechanism. Thus, including all patients under the umbrella of AR is not scientifically correct in this context. Needs justification.

Response: One of the weakness and (and concomitantly a strength) of this study is that it is observational. As such rhinitis is characterized by participating physicians in their usual way. We agree completely that in order to make a definitive definition of AR, SPT and/or specific IgE measurement should be done. However, in real life many physicians simply don’t do this either because they (1) feel they do not need to (e.g. are happy to make a diagnosis of AR based purely on clinical presentation – particularly the presence of clear rhinorrhoea as per ARIA 2008)³ and/or (2) they do not have the resource available. We have altered the wording in the methods section (page 7, line 133) to show that patients must have had a documented allergy. Skin prick test or specific IgE may have been done by the participating physician or previously. We have also made it clear that some patients in the unknown group may have had NAR (page 16, line 255).

At this point, I should suppose that the definition of PAR, SAR, SAR+PAR were supported by an IgE sensitization diagnosis, but this is not clearly described in the text. Were skin test and/or IgE assay performed in all patients?

Response: please see response above

Page 8, line 37. MP-AzeFlu use in clinical practice. First sentence "Information on patient demographics, clinical symptoms, and previous AR treatments in the last calendar year (prior to MP-AzeFlu prescription) was documented by physicians". It is not clear if the study was truly prospective or if data were collected also retrospectively. If this is a prospective study, the Authors must clearly state that patients were enrolled starting 1 year prior to the prescription of the product. If not, the study is not prospective.

Response: Thank you, yes completely agree – while patients were included in a prospective manner we asked about patient characteristics relevant at that time including those generated in the preceding year, so you are right the data was collected in parts retrospectively. We have changed that (page 2, line 28; page 7, line 118).

How were data collected? Agreed database? Questionnaire? Interviews? Clinical records' review? Please specify this aspect.

Response: Data were recorded by physicians in a pre-specified electronic case report form (eCRF) (Trium Analysis Online GmbH) based on the investigators patient files and information obtained throughout the patient visit. This information was included on page 9, line 165.

How were the involved physicians enrolled? How were they distributed (tertiary/secondary, specialists/GPs etc)

Response: Physicians who were usually involved in AR management participated in this study and included secondary care ear-nose-and-throat (ENT) specialists, immunologists and allergists. These physicians were identified by an independent agency (pH Associated, UK) (page 8, line 150). Another formal problem is that rhinitis is classified for severity according to ARIA criteria, but for duration according to the classic distribution (PAR/SAR).

Response: The decision was taken to categorize patients according to the traditional classification system (i.e. PAR/SAR) since it was considered that many physicians would be unaware or unused to the intermittent/persistent ARIA classification.³ Moreover, the SAR/PAR classification is also reflected in Dymista's labelled indication⁴ and it was intended to explore which kind of patients actually would be seen as target group in practice. Finally, the four severity criteria as outlined in the 2008 ARIA guideline³ may be used both for SAR/PAR and intermittent/persistent AR classifications [communication from Prof Jean Bousquet].

According to Fig 1, one third of patients had no physician visit in the previous year. How was the drug prescription assessed in those patients?

Response: X-axis of Figure 1 refers to number of previous doctor visits in the current year – i.e. previous to the current visit.

Reviewer: 3

Reviewer Name: Caral Irani

Institution and Country: Hotel Dieu de France hospital, St Joseph University, Beirut, Lebanon

Please state any competing interests: None declared

Please leave your comments for the authors below

The authors have as main objective to see the control of AR prior to Azelastine. The results for that regard are clear. [Our aims were not to see control of AR prior to Azelastine, but rather

to characterise the type of patient prescribed MP-AzeFlu (Dymista) in real-life in the UK and physicians' reasons for prescribing it and (ii) to quantify the personal and societal burden of AR in the UK prior to MP-AzeFlu prescription.]

Discussion of "Dymista" reasons for prescription need to be better explored and discussed. It seems that the study could have been done even without mentioning the introduction of "Dymista"

Response: Physicians were presented with 3 possible tick boxes for their reason to prescribe Dymista: "other therapies were not sufficient in the past", "other therapies were not considered to be sufficient to treat acute symptoms", or "other". We now have explored this result in discussion, and contextualized it with reference to patients' previous treatment(s) (pg 17, line 278).

Mentioning of Dymista (MP-AzeFlu) is necessary since our aim is to characterize the type of patient prescribed this new class of AR medication in the UK. At the time of this study, Dymista was not included in any AR guideline and so it was of interest to us how UK physicians would interpret the Dymista indication as written in the SPC. Since then, of course we have the updated MACVIA ARIA AR CDSS which recommends Dymista or INS for those with intermittent or persistent disease and a VAS Score >5/10 cm.²

Reviewer: 4

Reviewer Name: AYSE BACCIOGLU

Institution and Country: KIRIKKALE UNIVERSITY FACULTY OF MEDICINE, DEPARTMENT OF PULMONARY DISEASES, DIVISION OF IMMUNOLOGY AND ALLERGY, KIRIKKALE, TURKEY

Please state any competing interests: None declared

Please leave your comments for the authors below

1. There are some minor spelling mistakes. For example; abstract- Conclusion: many patients, introduction: line6- ...(AR), 1 2....

Response: Thank you. We have checked for spelling mistakes.

2. Methods: detailed prescription about MP-AzeFlu may be added (the duration, indications, adverse events....).

Response: We have included specific indication for MP-AzeFlu and refer the reader to the full SPC which is available online (page 7, line 120).⁴

3. Study design is confusing. Even it is prospective, all the data is collected retrospectively. It was written that "patients were treated with MP-AzeFlu". So did all patients use MP-azeFlu? Where are the results of the drug use? Whom were most recovered group?

Response: Thank you – we have clarified the study design. In fact MP-AzeFlu patients have been included prospectively and patient characteristics known at that time were to be captured to assess which patients would indeed get this medication. Hence, you are right, all patients got MP-AzeFlu. Further note that in 5 other countries where similar studies have been conducted effectiveness was assessed prospectively.¹ However, this was not possible in a pure observational study in the UK since the use of a VAS was considered as an intervention. However, we do describe results (in the discussion section, page 18, line 305) from other prospective studies of identical design carried out in Sweden, Norway, Denmark, Germany and Romania which did assess effectiveness and showed that MP-AzeFlu treatment was associated with a rapid, statistically significant, clinically relevant and sustained VAS score reduction. The MACVIA ARIA VAS score cut-off of 5/10 cm was reached by patients in a matter of days.¹

4. Is this a company sponsored study?

Response: Yes this study was sponsored by Meda Pharma, UK. That information is included in the acknowledgements section (page 20, line 355).

5. As I understand this is not a study showing the effectiveness of drug, so we don't know if MP-AzeFlu is more effective than nasal steroid or oral antihistamine. So the last sentence of the conclusion "A more effective treatment option, like MP-AzeFlu, should improve AR control and reduce costs associated with its management" is only an assumption, and should be discarded or replaced to a mild compliment about MP-Azeflu.

Response: Thank you . We have amended this sentence to more accurately reflect the aims of the study, i.e.:

In the UK, MP-AzeFlu was prescribed for individuals (≥ 12 years) with moderate/severe AR irrespective of (i) previous AR treatment (mono or multiple), (ii) previous or likely treatment failure, (iii) phenotype, (iv) number of previous physician visits for AR and (v) for the relief of both acute symptoms and in anticipation of allergen exposure (page 19, line 335).

Since the efficacy of MP-AzeFlu has been established in both RCTs^{5 6} and in real-life¹, we have amended the last sentence to read: A more effective treatment option, like MP-AzeFlu, with established superiority over intranasal corticosteroid and anti-histamine monotherapy, should improve AR control (page 19, line 338). We have removed potential for cost-saving.

6. Some results comparing azelastine versus nasal steroid in AR/NAR and the for a combination drug may be added (Kalpaklıoğlu, A.F., Baççioğlu Kavut, A., "Comparison of azelastine versus triamcinolone nasal spray in allergic and nonallergic rhinitis", *Am J Rhinol Allergy*, 24, 29-33 (2010).

Response: Thank you, but comparison of AZE vs INS is beyond the scope of this study. A study comparing AZE and FP has previously been published by Carr and colleagues,⁷ showing AZE provides comparable symptom control to FP in patients with moderate/severe SAR. Our aim was to characterize patients who received Dymista (not AZE) in the UK.

References

1. Klimek L, Bachert, C., Stjarne, P., Dollner, R., Larsen, P., Haahr, P., Agache, I., Scadding, G., Price, D. MP-AzeFlu provides rapid and effective allergic rhinitis control in real-life: a pan-European study. *Allergy Asthma Proc* 2016;37(5):376.
2. Bousquet J, Schunemann H, Arnavielhe S, et al. MACVIA clinical decision algorithm in allergic rhinitis in adolescents and adults. *J Allergy Clin Immunol* 2016;138:367-74.
3. Bousquet J, Khaltaev N, Cruz AA, et al. Allergic Rhinitis and its Impact on Asthma (ARIA) 2008 update (in collaboration with the World Health Organization, GA(2)LEN and AllerGen). *Allergy* 2008;63 Suppl 86:8-160. doi: 10.1111/j.1398-9995.2007.01620.x
4. Dymista. Summary of Product Characteristics [Available from: <https://www.medicines.org.uk/emc/medicine/27579>.
5. Carr W, Bernstein J, Lieberman P, et al. A novel intranasal therapy of azelastine with fluticasone for the treatment of allergic rhinitis. *J Allergy Clin Immunol* 2012;129(5):1282-89 e10. doi: 10.1016/j.jaci.2012.01.077
6. Meltzer E, Ratner P, Bachert C, et al. Clinically relevant effect of a new intranasal therapy (MP29-02) in allergic rhinitis assessed by responder analysis. *Int Arch Allergy Immunol* 2013;161(4):369-77. doi: 10.1159/000351404
7. Carr WW, Ratner P, Munzel U, et al. Comparison of intranasal azelastine to intranasal fluticasone propionate for symptom control in moderate-to-severe seasonal allergic rhinitis. *Allergy Asthma Proc* 2012;33(6):450-8. doi: 10.2500/aap.2012.33.3626

VERSION 2 – REVIEW

REVIEWER	Giovanni Passalacqua Allergy and Respiratory Diseases University of Genoa - Italy
REVIEW RETURNED	31-Jan-2017

GENERAL COMMENTS	The Authors have made a commendable effort in amending /ammeliating their MS, and most of my previous comments were satisfactorily addressed. I remain in the opinion that the word “allergic rhinitis” in the title is not fully appropriate since, as acknowledged by the Authors in their responses, a detailed pathogenic diagnosis was not done in all cases. I acknowledge that this reflects the real-life setting situation, but does not justify the claim of the title. The same applies to the use of terms PAR and SAR. I would suggest to reword the title, avoiding the term “allergic”, that is not clearly documented. Again, in the “limitations” I would suggest to underline that no direct head-to-head comparison between the investigated treatment and others available was made. Thus, no superiority or equivalence can be directly deduced in this context.
---

REVIEWER	AYSE BACCIOGLU Kirikkale Univ Medical Faculty Department of Pulmonary Diseases Division of Immunology and Allergy
REVIEW RETURNED	31-Jan-2017

GENERAL COMMENTS	I have no additional reviews, Its a honor to review Prof Scadding's paper. Have a nice citations.
--

VERSION 2 – AUTHOR RESPONSE

Reviewer: 2

Reviewer Name: Giovanni Passalacqua

Institution and Country: Allergy and Respiratory Diseases, University of Genoa - Italy

Please state any competing interests: None to declare

Please leave your comments for the authors below

The Authors have made a commendable effort in amending /ammeliating their MS, and most of my previous comments were satisfactorily addressed. Thank you

I remain in the opinion that the word “allergic rhinitis” in the title is not fully appropriate since, as acknowledged by the Authors in their responses, a detailed pathogenic diagnosis was not done in all cases. I acknowledge that this reflects the real-life setting situation, but does not justify the claim of the title.

We have removed ‘allergic’ from the title

The same applies to the use of terms PAR and SAR. I would suggest to reword the title, avoiding the term “allergic”, that is not clearly documented. We have removed ‘allergic’ from the title

Again, in the “limitations” I would suggest to underline that no direct head-to-head comparison between the investigated treatment and others available was made. Thus, no superiority or equivalence can be directly deduced in this context.

Thank you. We have done so (page 18, line 322)

Reviewer: 4

Reviewer Name: AYSE BACCIOGLU

Institution and Country: Kirikkale Univ Medical Faculty, Department of Pulmonary Diseases, Division of Immunology and Allergy

Please state any competing interests: None declared

Please leave your comments for the authors below

I have no additional reviews, Its a honor to review Prof Scadding's paper. Have a nice citations.
Thank you for your kind comments and for taking the time to review our manuscript.

VERSION 3 – REVIEW

REVIEWER	Giovanni Passalacqua Allergy and Respiratory Diseases IRCCS San Martino-IST-University of Genoa Italy
REVIEW RETURNED	20-Feb-2017

GENERAL COMMENTS	The Authors have amended title, commentaries and conclusions, avoiding the term "allergic" rhinitis, not documented. For the remaining, the study represents the general clinical practice context. My comments were satisfactorily addressed.
--